# ScaleTraversal: Creating Multi-Scale Biomedical Animation with Limited Hardware Resources

## ABSTRACT

We design ScaleTraversal, an interactive tool for creating multi-scale 3D demonstration animations with limited resources for users who are unavailable to access high performance machines such as clusters or super computers. It is challenging to create 3D demonstration animations for multi-scale data. First, it is challenging to strike a balance between flexibility and user friendliness to design the user interface in customizing demonstration animations. Second, the multi-scale biomedical data is often characterized as large-size so that a commonly-used desktop PC is hard to handle. We design an interactive bi-functional user interface to create multi-scale biomedical demonstration animations intuitively. It fully utilizes the strengths of graphical interface's user friendliness and textual interface's flexibility, which enables users to customize demonstration animations from macro-scales to meso- and micro-scales. Furthermore, we design four scale-based memory management strategies to solve the challenging issues presented in multi-scale data. They are streaming data processing strategy, scale level data prefetching strategy, memory utilization strategy, and GPU acceleration strategy for rendering. Finally, we conduct both quantitative evaluation and qualitative evaluation to demonstrate the efficiency, expressiveness and usability of ScaleTraversal.

## CCS CONCEPTS

• **Human-centered computing** → **User interface programming**; *User interface toolkits.*

## KEYWORDS

interactive experience tool, multi-scale multimedia data, storytelling animation

## 1 INTRODUCTION

Most of the existing data-driven illustration tools for biomedical data only provide a visual presentation at a single scale level [18]. For example, tissue scale such as Facetto [29], fiber scale ($10^{-2}m$) such as ConnectomeExplorer [5], cell scale ($10^{-4}m \sim 10^{-5}m$) such as Abstractocyte [39], neuron scale ($10^{-5}m$) such as NeuroLines [3], neuronal structure representation [7] and neurobiology data presentation [48], and DNA scale ($10^{-9}m$) such as nucleic acids [31] (RNA and DNA) and interaction predictions of sRNA-mRNA [45]. Biomedical data, however, consists of various scales of data which

*ACM MM, 2024, Melbourne, Australia*
© 2024 Copyright held by the owner/author(s). Publication rights licensed to ACM.
ACM ISBN 978-x-xxxx-xxxx-x/YY/MM
https://doi.org/10.1145/nnnnnnn.nnnnnnn

often range from several orders of magnitude, e.g., a multi-scale brain data often comprise structures include brain envelop ($10^{-1}m$), encephalic region ($10^{-2}m$), fiber ($10^{-3}m$), cortical column ($10^{-4}m$), neuron ($10^{-5}m$), chromosome ($10^{-7}m$), DNA or RNA ($10^{-9}m$).

The problem thus arises of how to illustrate the multi-scale biomedical data in multi-angle and multi-aspect. 3D demonstration animation is an effective approach to reveal the multi-angle and multi-aspect information presented in the scientific data [32]. It can be used in knowledge popularization, peer-expert sharing, or peer-expert discussion. There are two motivations as well as some corresponding challenges in creating and customizing data animation for multi-scale biomedical data:

First, it is of great significance to help users comprehend and explore multi-scale information presented in the data [18]. However, it poses a great challenge for illustrating static visual presentations for biomedical data, not to mention the intelligent creation and customization of 3D demonstration animations. It needs to solve the challenging issues both in 3D visual designs, visual presentation customizations, and animation customizations [32]. Visual presentation customization includes customizing visual styles of point, line and surface and other components, while animation customization includes the selections of viewpoints, transition styles and shot change styles. Furthermore, generating data-driven animations and keeping precise controls are challenging because generating slight, gentle and elegant transitions within the animation is difficult to achieve by manual GUI interaction [32]. Besides, as the virtual camera gradually focuses from the cerebral cortex of the brain envelop scale to the chromosome scale or even DNA scale, macro-scale objects in the virtual camera will quickly disappear in the viewpoint. The large difference of orders of magnitude across scales pose a significant challenge in smooth transition in the final demonstration animation.

Second, the multi-scale often results in the data to be large-size. The animations should cover all the scales of the data they have provided and the gradual transitions between scales should be smooth. The "large-size" in this paper refers to the data size is too large to handle by a hardware equipment with limited resources, e.g., a desktop PC for common users. It is impossible for the users to handle multiple scales of data simultaneously, especially for the domain experts who are often unavailable to access high performance machines such as clusters or super computers. The "limited resource" in this paper refers to the hardware configuration used in generating demonstration animations by using the traditional pipelined animation rendering method [32] will result in application crash (out of memory).

In this paper, we present a novel interactive data animation building tool named ScaleTraversal to address the above-mentioned challenges. ScaleTraversal enables users to customize multi-scale animations by easy and simple GUI operations like dragging, dropping and scrolling. It adopts simple and embedded domain-specific

language (DSL), allowing users to customize complex presentation styles and animation transitions flexibly. DSLs are languages that are designed for specific domains and decrease the learning curves for domain experts compared with general programming languages [32, 47]. In ScaleTraversal, the customized animation clips such as camera rotating around a given "encephalic region/object" within 30 seconds with a constant-speed, achieving gentle and elegant shot changing by transiting and zooming-in simultaneously when the camera travels from a large "encephalic region" to another smaller one, and specifying an accurate animation time and transition time for a specific animation clip. All of these clips are difficult to manually customized by users via the traditional devices like mouse and keyboard. To strike a balance between flexibility and user-friendliness, we use GUI primarily for user-friendliness and combine simple and embedded DSL codes to improve flexibility. All the statements of the DSL codes can be auto-generated by GUI selections using icon buttons. Users just need to change the parameters of each statement. The identifiers are not allowed changed by users if they have chosen a script code unless they want to choose a new one. Furthermore, we design four scale-based memory management strategies to address the challenges in the scenario of limited hardware resources.

The main contributions of this paper can be summarized as follows:

- We design an interactive bi-functional user interface to customize multi-scale biomedical demonstration animations intuitively. It consists of a graphical (i,e, GUI controls) and a textual grammar (simple DSL codes). They are fully utilized to strengths of GUI's user-friendliness and textual grammar's flexibility.
- We design four multi-scale data acceleration strategies with limited hardware resources for users who are unavailable to access high performance machines, including a streaming data processing strategy, a scale-based data prefetching strategy, a memory utilization strategy and a GPU rendering acceleration strategy.

## 2 BACKGROUND AND RELATED WORK

### 2.1 Muti-scale Data Presentation Tools and Systems

There are many technique challenges in designing multi-scale tools or systems due to the significant scale differences between scales. We summarize and categorize the multi-scale tools into three types according to the application scenarios, i.e., rendering, interface design, simulation and analysis, image data processing.

**Multi-scale interface design**. A multi-scale text retrieval interface [26] was presented to provide a detail+context views of documents information. The interface allows users to compare small multiples in a detailed view. A 3D geological hazard user interface [35] was also designed to integrate intuitive interaction and 3D visual presentations for multi-scale spatially referenced data. Furthermore, a multi-scale flight presentation interface [41] was designed to verify whether eye height would affect the user's perception of object size.

Recently, some impressive multi-scale data presentation tools were designed. For example, a multi-scale procedural model [28]

was built to simulate complex characteristics of a biological process about microtubule dynamics for measured data. A method named ScaleTrotter [18] was proposed to bridge several orders of magnitude in scale, allow observers to gradually and smoothly observing the DNA composed of atoms from the nucleus. A tool named Dynamic Scene Graph [4] was designed to address the scaling and navigation issues in multi-scale universe simulation data. It assigns frame of reference dynamically to achieve the highest possible numerical precision for objects rendering in a scene graph.

**Multi-scale biomedical simulation data analysis**. Stochastic multi-scale modeling of biological systems needed the use of mathematical models that can describe different levels of scales. A new parallel computational paradigm [16, 23] was proposed to achieve multi-scale simulations in brain blood flow data. They coupled several parallel codes to form a multi-scale solver and each code was built based on different mathematical models. Blackett et al. [6] described some models to simulate a heart ventricle by solving multi-scale and multi-physics issues. The simulations that run on peta-scale supercomputers pose a great challenge for scientists to analyze their data remotely. To address this computational challenge on multi-scale data analysis, Ahrens et al. [2] provided a prioritized, streaming and multi-resolution architecture. Regarding the flow simulation in fluid dynamics, Nguyen et al. [40] designed a data presentation framework to divide large-scale structures into dense, small-scale structures.

Most of the existing multi-scale data presentation tools focus on static rendering instead of demonstration animation generation and customization.

### 2.2 DSL-Based User Interface Design

DSLs are languages that are designed for specific domains and decrease the learning curves for domain experts compared with general programming languages [32, 47]. Compared with high-level interfaces [43], DSLs provide brevity rather than generality. On the basis of three fundamental dimensions of DSLs, i.e., abstract syntax, concrete syntax and semantics, DSL-based interactive tools can be categorized into tree types according to different perspectives [47], they are external or internal DSLs, textual or graphical DSLs, and DS"X"L.

**External or Internal DSLs.** An external DSL is a language that's parsed independently of the host general purpose language: good examples include regular expressions and CSS. In the domain of scientific data processing and image analysis, Kindlmann et al. [27] proposed Diderot, a typical example of *External DSLs*, which decreases obscurity caused by low-level implementation by providing high-level mathematical notations.

**Textual or Graphical DSLs.** Concrete syntax map language structures to some notations categorized into textual and graphical, which brings about the *Textual DSLs* and the *Graphical DSLs*.

DSLit, a method driven by model was proposed by Cosentino et al. [12] to avoid high-cost while designing *Textual DSLs*. Halide [42] introduced an *Textual DSLs* based on C++, provides high performance and optimizes the program. In the domain of volume data rendering, Delite [9] and Vivaldi [10] were designed for heterogeneous systems. Nevertheless, the cost of learning and using is high

for users have to comprehend specific declarations and semantic rules [46].

**DS"X"L.** Numerous DSLs provide different glyphs according to the exact domain. These techniques are named DS"X"L normally. The related work can be categorized into three types [47], i.e., DSVL (Domain-specific Visualization Language), DSML (Domain-specific Modeling Language) [13], DSEL (Domain-specific Embedded Language) [22]. (1) **DSVL**. As the need to satisfy the need in engineering fields, a set of visualization methods exploring seismic slice and seismic volume data [33, 34] are typical examples of DSVL. (2) **DSML**. DSML allows creating behavioral models of 3D objects in a specific modeling environment. Choi et al. [10] designed Vivaldi with the help of shape grammar, achieving multiple parallel processing strategies without requiring much knowledge of it. (3) **DSEL**. Michael et al. [8] proposed ProtoVis based on JavaScript-based *Embedded DSLs*.

### 2.3 Context and Relationship

There are little literatures on customizing multi-scale 3D demonstration animations for biomedical data. The most relevant work includes a latest work named ScaleTrotter [18], the extension work named Multiscale Unfolding [17], and a multi-scale procedural model [28]. ScaleTrotter is a multi-scale data demonstration tool to present biological meso-scale genome data, which allows generating visual traversals across scales by roaming between 2D and 3D renderings. Another multi-scale tool named Unfolding [17] was designed to illustratively render multiple hierarchical scales of DNA in a single view by spatial folding. The multi-scale procedural model [28] was designed to simulate complex characteristics of a biological process about microtubule dynamics for measured data. Dynamic Scene Graph [4] focuses more on avoiding precision-related rendering artifacts and the seamless adaption schemes in stereoscopic rendering, while the proposed ScaleTraversal focuses on user interface design to customize multi-scale biomedical animations, and data management strategies for accelerations. Additionally, a framework named multi-scale views [20] was designed to provide fcocus+context effect and artistic rendering to illustrate multi-scale data into a single image instead of an animation. The streamline-based ray generation often produces distortions between scales and even within a scale in the final image. The main differences between ScaleTraversal and the most relevant latest work can be summarized as follows:

(1) **Just focusing on animation generation of genome scale data (i.e., nucleus and DNA scale) v.s. focusing on data-driven animation generation and customization ranging from envelop scale (macro-scale) to DNA scale**: Most of the existing tools are designed to generate multi-scale genome data animation, ScaleTraversal is to generate and customize multi-scale animations from macro-scales to meso- and micro-scales by embedding flexible short DSLs.

(2) **Few contributions on data management v.s. Multi-scale data management and GPU rendering even on desktop PCs**: Multi-scale biological or biomedical data are often characterized as large-scale data size due to their multiple scales of data. It is challenging to customize multi-scale 3D data-driven animations of

biomedical data on a desktop PC for users who are unavailable to access high performance machine such as cluster or super computer. Few existing approaches in this aspect involve contribution on multi-scale data accelerations. ScaleTraversal contributes to several acceleration strategies such as buffer management, data prefetching across scales and GPU rendering improvements to address both multi-scale and large-scale challenges. It enables users to customizing multi-scale animations with limited hardware resources.

(3) **Couple 2D and 3D presentation v.s. Support 3D animation throughout**: ScaleTrotter [18] uses 2D presentation in a specific data scale and 3D transitions between two scales. However, ScaleTraversal supports 3D animation throughout due to the acceleration strategies on data management and GPU rendering.

## 3 DESIGN RATIONALE

In this section, we describe the design rationale in terms of data characteristics, design goals, design considerations and design details.

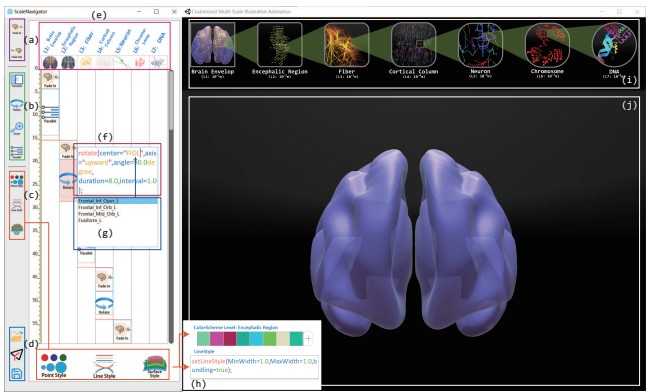

**Figure 1: The interface design of ScaleTraversal, it includes: a group of controls that can be drag to customize the fade in or fade out of scales (a), animation transitions (b), and basic visual styles for different types of objects (c and h); buttons of loading previously saved scripts or the scripts shared by peer experts (d); all of the data scales (e); a popped-up text editor that is mapped to a clickable GUI control, where users are just allowed to modify the parameters of DSL functions (f); auto completer that match similar words when users edit the parameter values (g); a group of scale previews indicate which scale virtual camera enters and where the virtual camera locates in all the previous scales (i), and the animation view (j).**

### 3.1 Characteristics of Multi-scale Biomedical Data

There are two challenges to customize data animations for multi-scale biomedical data on a desktop PC, as summarized in section Introduction: i.e., multi-scale challenge and the derived large-size challenge for desktop PCs. The envelope of the cerebral cortex is about $1.5 \times 10^{-1} m$, and the neurons inside the brain are only about $2.0 \times 10^{-3} m$, while the DNA is only $2.0 \times 10^{-9} m$. There are eight orders of magnitude in scale between the brain envelope and DNA.

When the virtual camera gradually travels from macro-scale to micro-scale, the errors which are derived from the number limits of data serialization may cause the camera shake severely when generating animations. That means the large order difference of magnitude across scales poses a significant challenge in generating elegant and smooth data animations automatically.

In addition, there are a huge number of objects in the tissues at all levels of scales. Regarding the typical biomedical data we obtained from the domain experts, the number of elements that should be rendered had exceeded 160,000 at L1 scale of cerebral cortex (L1: brain envelope), and at the nerve fiber level (L3: fiber), experts provided more than 10 GB of data on the probability of connections between different voxels in the brain. Any two of these voxels can be connected to each other with a line of thickness corresponding to the connection probability. At the neuron level (L5: neuron), each neuron contains between 150,000 and 300,000 elements. A normal brain has about 86 billion neurons, and each neuron has to establish nearly 10,000 links.

## 3.2 Design Goals and Design Considerations

We consulted four domain experts in biological study about the requirements, who also provided us with the biomedical data. The significant issues they are concerned about include three aspects, we summarized them as three design goals:

- G1: design an interface to create multi-scale biomedical demonstration animations intuitively.
- G2: enable users to flexibly customize animations from overview to details.
- G3: enable users to build animations with all the scales of data under limited hardware resources.

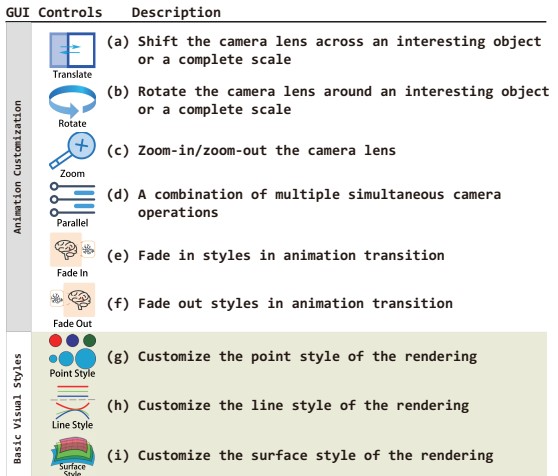

**Figure 2: GUI controls designed for data animation customizations. Users can click and drag the GUI controls on the control bar into the animation track.**

Users can either directly explore the customized demonstration animations in real-time by using ScaleTraversal or demonstrate the output data-driven animation videos in **knowledge popularization** in their talks, **peer-expert sharing**, or **peer-expert**

**discussion**. All the design details are derived from the three design goals.

## 3.3 Design Details

**Interface overview**. We design a linked view in ScaleTraversal for domain experts to customize the multi-scale data animations intuitively with a visual preview (G1), and further design a waterfall-like customization interface, which follows the design in the video editing software Adobe Premiere, as shown in Figure 1. The data-driven demonstration animations will be automatically generated after the customization is completed by the operations in Figure 1 (a-h).

**Animation customization**. Users can select and drag different GUI controls by simple GUI operations to customize the data animation across different scales of data. The current animation progress will be shown in the waterfall-like interface. Different GUI controls represent different customization operations or animation styles.

The DSL script codes are easy to be edited because the domain experts just need to choose the parameters from a UI list instead of memorizing all the identifiers and parameters (G2). All the GUI controls in Figure 1 (a-c) can be clicked or dragged to add their corresponding DSL scripts into the animation track in Figure 1 (e). To further decrease the learning curve and improve the usability of ScaleTraversal, all the tissue names related to the animation presentation will be listed in a GUI list control, which will be popped up close to the corresponding parameter positions in the interface, as shown in Figure 1 (f). Users can select them by GUI operations by visual cues and prefix matching algorithms (G1 and G2).

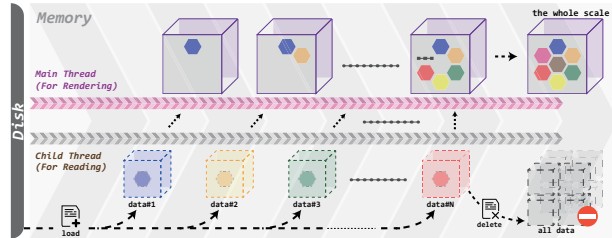

**Figure 3: Illustration of streaming data processing strategy.**

In the animation customization stage, users are allowed to translate the camera along a given direction specified by assigning the parameters of the GUI control as shown in Figure 2 (a). All the GUI controls would be directly mapped to a short DSL script codes. The camera rotation around an object in a given scale can be customized by using the GUI control as shown in Figure 2 (b). For example, ScaleTraversal allows to customize the animation by rotating the virtual camera around the frontal lobe or the parietal lobe. Besides, users can also zoom-in or zoom-out the virtual camera by using the corresponding GUI control, as shown in Figure 2 (c). Furthermore, some complex customization requirements in animation generation can be fulfilled by "parallel" operations, which are designed to combine multiple single operations like translating, rotating and zooming into a "parallel" script statement. Our tool will automatically zoom-in the animation when it travels from a large object to

a smaller one. The zoom-in factor depends on the size difference of them (G1). Finally, users can customize different transition effects about fade-in and fade-out between two scales. The corresponding GUI controls are shown in Figure 2 (e) and Figure 2 (f), respectively.

We built a light-weight compiler following the DSL framework design in IGScript [32]. It is exploited to analyze the script codes and further mapped them to visual presentation codes and automatic virtual camera operations. For more details about the design of the compiler and DSL codes, please refer to the **Appendix** file.

**Visual style customization**. (1) Point style customization. Users can specify colors and transparencies for point objects and spherical objects. The color array can be assigned to different clusters of points. For example, points in an identical cluster will be assigned with an identical color, or the points in an identical encephalic region will be assigned with the identical color. The point style customization script codes can be popped up by clicking the GUI control as shown in Figure 2 (g). (2) Line style customization. Similarly, the line style can be customized by clicking the GUI control as shown in Figure 2 (f). Users can modify the parameters of the script codes in the popped-up DSL script editor like in Figure 1 (h). Besides, users are allowed to define whether need to bundle the lines at a given data scale. When there are a large number of lines in a local region, it often induces severe visual clutter or 3D occlusion. If the bundling effect is turned on, all the lines will be bundled by an edge bundling algorithm named FDEB [19]. All the line bundling should make sense according to the domain knowledge. (3) Surface style customization. To help users distinguish materials of different tissue faces and enhance their realism, we provide functions to customize the glossiness and metallicity of the surface (Figure 2 (i)). ScaleTraversal uses different lighting models or different lighting parameters to achieve the effects.

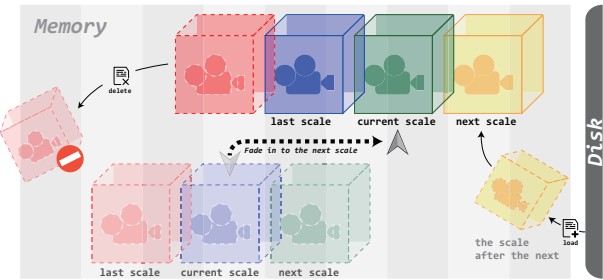

**Figure 4: We design a prefetching and a dynamic dealloca-tion strategy to read and render data during the animation generation, and release the least recently used scale of data from the memory.**

## 4 SCALE-BASED ACCELERATION STRATEGIES

We design four scale-based acceleration strategies under limited hardware resources for users who are unavailable to access high performance machines, including a streaming data processing strategy, a scale-based data prefetching strategy, a memory utilization strategy and a GPU rendering acceleration strategy (G3).

**Streaming data processing strategy**. We design a read-render data processing strategy to avoid long-time blank when ScaleTraversal starts to read data, because it often takes too much time to initialize the data loader and renderer, load all scales of data, and render the first scale of data specified by users (G3). Generally, when faced with a large amount of data regarding limited hardware resources, e.g., a common desktop PC, reading and rendering all at once leads to a dramatic increase in memory consumption, and rendering all the data in a short period of time leads to extremely low FPS. Our approach supports the synchronization of data reading and rendering in different threads. First, the data reading thread reads a small amount of data from the first scale into the buffer. When the data in the buffer accumulates to a certain size, the rendering thread will fetch that data from the buffer and render it promptly, while the reading thread is still reading the data and pushing them into the buffer, as shown in Figure 3. The streaming data processing strategy makes the rendered objects appear in the animation one by one to avoid long-time blank and further increases the FPS of the animation.

**Scale-based prefetching and dynamic memory dealloca-tion**. The streaming read-render data processing strategy is useful to increase the FPS when starting an animation. However, we still need better data management strategies to further increase the overall FPS. We design a scale-based prefetching strategy and a dynamic memory deallocation strategy (G3). For example, we can customize an animation from the scale of encephalic region to the scale of fiber. When the animation of encephalic region starts, the fiber visualizations are completely invisible for us. At this time, a prefetching thread will start to read the next scale of data. It can effectively accelerate the scale-based data reading. When the animations enter the fiber scale, the prefetching thread will start to load the data of the next scale, e.g., the cortical column scale. Furthermore, we can release the first scale of data (encephalic region) when the animation starts to enter the third scale level (cortical column), as shown in Figure 4. The scale-based prefetching strategy and the dynamic memory deallocation strategy can further help to increase the FPS of the overall animation.

**Super object bundling**. We created a super object combination strategy to save the data and state information of multiple objects in batch, while the CPU only needs to push the super object each time (G3). The strategy reduces the number of "Draw Calls" and further increases the FPS of animation rendering (G3). For more details about the super object design, please refer to the **Appendix** file.

## 5 ANIMATION EXAMPLES

We designed a light weight compiler following the compiler design (see Appendix) in IGScript [32]. It is used to translate the whole animation flowcharts (GUI controls and the embedded short DSL scripts) into DSL script codes. The whole script codes for an animation customized by users can be saved to disk and reused and re-edited in future. Meanwhile we develop a hub which acts as a data transfer station between the waterfall interface and the customized data animations. It is used to transfer script commands to the animation generation renderer, ensuring that the animation and the flowchart time are in tune when the animation is presented.

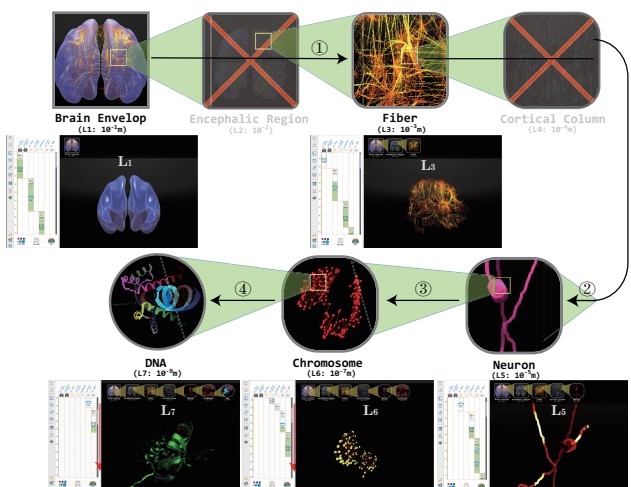

Figure 5: The first demonstration animation generated by five data scales. They are brain envelop scale (L1), fiber scale (L3), neuron scale (L5), chromosome scale (L6) and DNA scale (L7). Top row: the icons indicating which scale and where the current position the virtual camera are locating. Bottom row: animation snapshots for each customized data scale. The current animation progress is indicted in the waterfall-like interface simultaneously.

ScaleTraversal was built on a desktop PC with limited hardware resources, e.g., Windows 10 Home with an Intel i7-10875H 2.30 GHz CPU, 16 GB RAM, and NVIDIA RTX 2060 Graphics card. The waterfall-like flowchart is developed on Qt5, and the rendering system of ScaleTraversal is built using GPU programming. The compiled results will be transferred to the rendering system to generate data animations automatically. The rendering system is designed similar to an API library, which can be flexibly changed. More additional rendering functions can be added afterwards, which improve the scalability of the rendering system.

In our experiments, we have conducted three case examples to evaluate the expressiveness and usability of the customization feature of ScaleTraversal. The third case study is moved to the **Appendix**.

The first case demonstrates the ScaleTraversal which allows users to customize the data animation across several selected scales they are interested in instead of all of the scales. Five scales the user is interested in are selected, as shown in Figure 5. They are brain envelop scale (L1), fiber scale (L3), neuron scale (L5), chromosome scale (L6) and DNA scale (L7). The animation building will skip two scales, encephalic region scale (L2) and cortical column scale (L4), because they are not customized to present in the final animation in this case, as shown in the two grey icons with red crosses in Figure 5. The top icons in the animation space indicating which scale and where the current position the virtual camera is locating. The bottom sub-figures are the corresponding snapshots of the whole customization interface. The detailed animation progress will be updated in real-time in the waterfall-like customization interface.

The second case consists of nine scales including a scheme of scale return, as shown in Figure 6. The animation flowchart includes two parts, the first part is to demonstrate the left-brain from brain

envelop scale (L1) to fiber scale (L3), while the second part is to demonstrate the detailed process in the right-brain from brain envelop scale (L1) to DNA scale (L7).

It is simple and easy for domain experts to customize data animations by dragging the GUI control for each scale into the animation customization track, as shown in the left part of each snapshot in Figure 5. All the GUI controls about animation transition customization and animation duration customization can be added into the animation track of each scale. Besides, the scale-based even the object-based visual styles can be also customized by short DSL script codes in a popped up window close the corresponding GUI control in the animation track. Domain experts do not need to memorize the identifies. They are just required to change the parameter values for each statement or choose the parameter values from a GUI list. For more details about how to customize the three animation cases, please see the **supplementary video** of the submission.

## 6 EVALUATION

The results are evaluated on the above-mentioned desktop PC with limited resources.

| Metrics | Pipelined [32] | The proposed |
|---|---|---|
| Raw Data Size | 3.24 GB | 3.24 GB |
| Memory Consumption | Out-of-memory | 7.7 GB |
| #Mesh Triangles | 239.1 M | 239.1 M |
| Minimum FPS | Out-of-memory | 22.3 |

Table 1: The overall performance metrics of the proposed ScaleTraversal compared with the traditional scientific data animation generation approach [32], which use traditional pipelined mode to process all the data and result in application crash due to out-of-memory. The actually used memory size during the animation is much larger than the original data size due to a large number of objects will be produced and rendered.

### 6.1 Performance Evaluation

**Overall quantitative performance**. The overall performance results of ScaleTraversal are shown in Table 1. It can be found that the minimum FPS of ScaleTraversal is 22.3. The baseline method is a recent work on scientific data animation generation [32], which uses traditional pipelined mode to process all the data. It should be noted that the actually used memory size during the animation rendering is much larger than the original data size on the experimental equipment, because there are a large number of objects will be produced and then a large number of mesh triangles with various lighting effects should be rendered, which usually takes a large number of memory consumption. The baseline method ends with application crash due to the gap between its resource intensive requirement and the limited resource on the identical experimental equipment (the desktop PC).

**Qualitative evaluation of streaming data processing strategy**. We evaluated the streaming data processing strategy by comparing it with the traditional non-streaming scheme. The experimental conditions are totally identical for the two methods. In this control group test, we just used a subset of the data (918.9 MB)

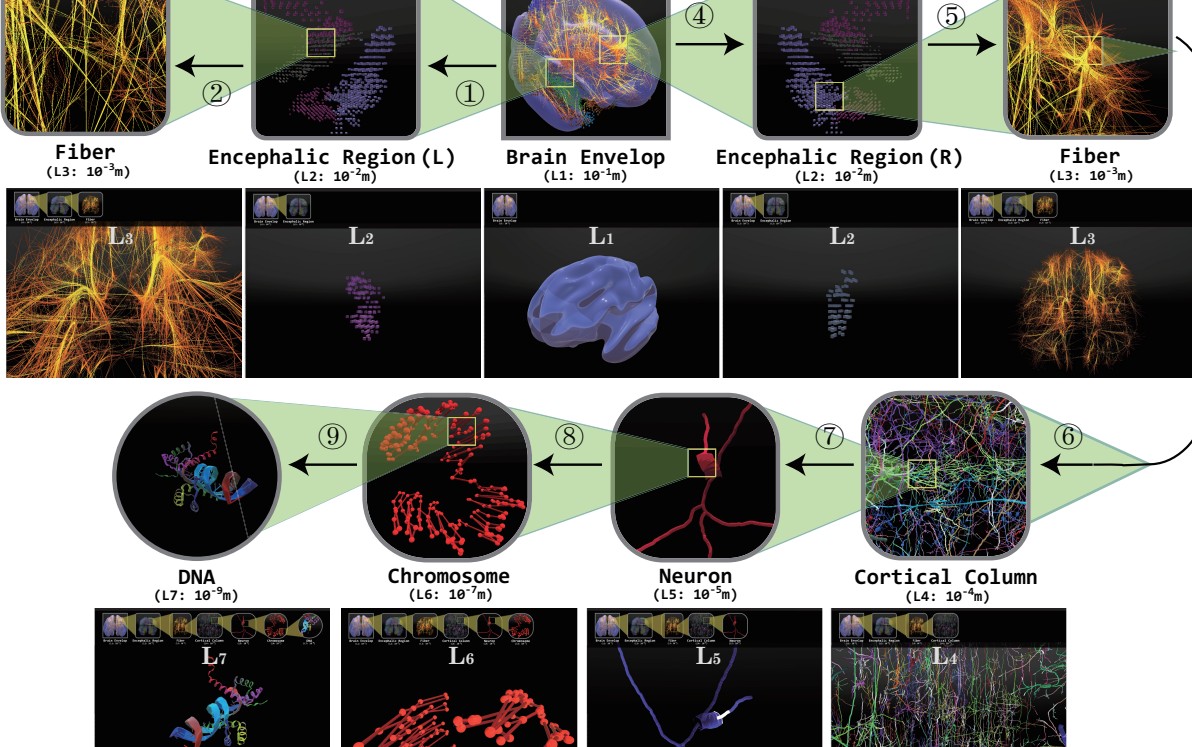

**Figure 6: The second demonstration animation is customized by using "scale return". The animation includes the demonstrations to show the "left-brain" and then the "right-brain". Top row: the icons indicating which scale and where the current position the virtual camera are locating. Bottom row: animation snapshots for each customized data scale.**

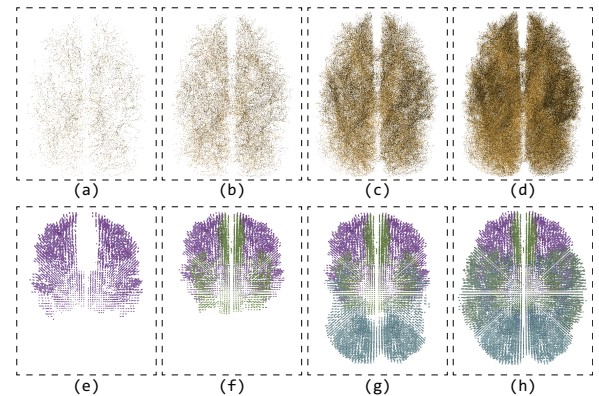

**Figure 7: The animation effect of streaming data processing strategy. Part of objects are displayed in the animation immediately after the corresponding data are loaded.**

to test the streaming strategy because the traditional method will crash if it loads all the scales of data. The traditional non-streaming scheme will take 25.49 seconds to show something in the animation due to the time-consuming data loading, object creating and object rendering. The qualitative result of the streaming strategy is shown in Figure 7.

**Quantitative evaluation of data prefetching strategy**. Similarly, we used two scales of data to test the prefetching strategy. The results are shown in Table 2. It can be found that the FPS with prefetching strategy up to 170 while the control group without prefetching strategy is just 25 FPS.

| Methods | No prefetching | Prefetching |
|---|---|---|
| Mesh Triangles | 215.9 M | 215.9 M |
| FPS | 25 | 170 |

**Table 2: Two scales of data are used to test the prefetching strategy, because the control method is out of memory and then crash if it loads all the data.**

**Quantitative evaluation of animation acceleration strategies**. We designed a super object combination strategy and a memory utilization strategy to accelerate the GPU rendering. The FPS with acceleration strategies is much larger than that without acceleration strategies (Table 3). We just used one scale of the data to evaluate the strategies due to the application of the control group will crash soon due to out-of-memory if loading all scales of the data.

| Mesh triangles | 271.3 K | | 11.4 M | |
|---|---|---|---|---|
| Method | No Acc | GPU Acc | No Acc | GPU Acc |
| Draw calls | 22603 | 94 | 9748 | 422 |
| FPS | 14.7 | 272 | 47 | 269 |
| Mem utilization | $\times$ | $\surd$ | $\times$ | $\surd$ |

**Table 3: Quantitative evaluation on two strategies including a GPU acceleration strategy (GPU Acc) by super object combination and a memory utilization strategy.**

## 6.2 Domain Expert Feedback

We asked for feedback from domain experts to further evaluate ScaleTraversal. Four domain experts who have engaged in brain study or neurology study are involved in the evaluation. They can either directly explore the customized demonstration animations in real-time on a desktop PC by using ScaleTraversal or demonstrate animation videos in **knowledge popularization** in their talks, **peer-expert sharing**, or **peer-expert discussion**. We received many feedback and suggestions. *"The animation is quite impressive actually"*, a feedback received from a well-known brain scientist when he first viewed the customized animation. We have been communicating with his team members for a long time about their requirements and new suggestions. After many iterations, the brain scientist said *"the biological side is quite reasonable"*. He really appreciates the flexible animations customized by our tool from the perspective of multi-scale and the smooth animation transitions between scales by providing context.

To get a more complete evaluation, we provide a questionnaire after their long-term use and back-and-forth improvements, where five point Likert scale is used (total scale $t = 5.00$). We received the questionnaire results: "The waterfall-like interface is *intuitive* and *easy* to customize animations" (Q1: $\mu = 4.00$, $t = 5.00$) and "the ScaleTraversal is efficient to customize animations" (Q2: $\mu = 3.67$). They said "the animation from macro-scale to micro-scale was helpful and effective" for them to understand the multi-scale biomedical data (Q4: $\mu = 4.00$). They pretty appreciated the GUI design embedded with textual codes, and they said the UI design was helpful and effective in animation generation and customization (Q5: $\mu = 4.33$).

Regarding the acceleration strategy effects, the domain experts said the streaming strategy (Q6: $\mu = 4.33$, Q7: $\mu = 4.00$) and the prefetching strategy (Q8: $\mu = 4.33$) are capable of improving the animation smoothness and fluency, after we showed the individual performance evaluation. We also consulted them about the overall feedback and the suggestions after the questionnaire. They said they like the documentary photography effect which is applied into the animation customization. Among the feedback, one of the senior experts who have studied multi-scale brain data for five years said *"I really appreciate the scale preview group on the top of the animation, which provides the overview progress and the current position of the camera in the macro perspective."* The overall feedback received from domain experts does not cover too much about the back-end acceleration strategies, because they pay more attention to the usability and expressiveness of the front-end interface and the customized animations.

Actually, the domain experts gave us many suggestions during the whole developing process of ScaleTraversal. For example, one of the experts said *"the flowchart in the waterfall-like interface is too long"*. Regarding this suggestion, we have improved the waterfall-like interface by adding an automatical slide bar to track the current progress. Besides, the flowchart can be zoom-in and zoom-out to make users better view the overview customization from the waterfall-like interface.

## 7 DISCUSSION AND CONCLUSION

We design a bi-functional user interface to customize multi-scale biomedical demonstration animations in this paper. Actually, there are another two alternative designs: **(1) Pure textual script interface**. It is a good option to obtain flexibility while sacrificing user-friendliness. At the start-up of this work, we have inquired many experts in various domains such as biologists, doctors, geologists, etc. Most of them said it is important to avoid them writing too many codes because they have gotten accustomed to or even relies strongly upon graphical interface like Mac OS, iOS, Windows, etc. We noticed the fact that the learning curves for them to study a new script language are really steep. **(2) Pure graphical interface**. In multi-scale data animation, it is hard for users to manually perform complex customizations as mentioned-above. For example, camera rotating around a given "encephalic region/object" within 10 seconds with a constant-speed, shot changing by transiting and zooming-in/zooming-out simultaneously when the camera travels across two biomedical objects with different sizes, and specifying an accurate time for fade-in and fade-out. Therefore, we use GUI primarily for user-friendliness and combine simple and embedded DSL codes to improve flexibility in ScaleTraversal.

It is impossible to process the large-size multi-scale data and render them simultaneously on a desktop PC. An alternative design to address the challenge is to re-sample the data at a much lower resolution. However, the down-sampling would result in data information loss in one/multiple of scales, and then would probably generate artifacts or distortions in the surface rendering for some objects.

Although ScaleTraversal is designed to customize multi-scale biomedical data animation, it is not tedious to extend it to other data with several orders of magnitude in scale, if given the corresponding scales of data. Because the customization interface is scalable to bind new 3D objects in the data, and the animation generator is extendable to use new 3D objects which the virtual camera will rotate around. Another significant step of the extension is to design the application-specific visualizations according to users' requirements.

In this paper, we introduce ScaleTraversal, an interactive tool for customizing multi-scale 3D demonstration animations (from macro-scales to meso- and micro-scales) with limited hardware resources. The target users are the domain experts who are unavailable to access high performance machines. It fully utilizes the strengths of graphical interface's user-friendliness and textual interface's flexibility. Furthermore, we designed four multi-scale data management strategies to make the animation as smooth as possible. Finally, we demonstrate the efficiency, expressiveness and usability of ScaleTraversal by performance test, case study and domain experts' feedback, respectively.

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
