# OpenReview forum: "ScaleTraversal: Creating Multi-Scale Biomedical Animation with Limited Hardware Resources"
_acmmm.org/ACMMM/2024/Conference — MM2024 Poster_

### Official Review · Reviewer_GwQz · 2024-05-19

**Rating:** 6
**Confidence:** 4

**Summary:**

This paper introduces ScaleTraversal, an interactive tool designed to create multi-scale 3D biomedical demonstration animations under limited hardware resources. The authors have developed a bi-functional user interface that combines the user-friendliness of a graphical interface with the flexibility of a textual interface, enabling users to intuitively customize demonstration animations from macro to meso and micro scales. Furthermore, the paper proposes four scale-based memory management strategies to address the challenges associated with processing multi-scale data. The efficiency, expressiveness, and usability of ScaleTraversal are demonstrated through quantitative and qualitative evaluations.

In my opinion, this is a high-quality paper with significant contributions that is well-suited for publication at ACM Multimedia. The authors have proposed a practical technical solution to address a key requirement in the biomedical field. The innovations are clear, the implementation is comprehensive, and the experiments are thorough. This work has the potential to make a positive impact in related domains. Therefore, I strongly recommend accepting this paper and consider it as a candidate for an Oral presentation.

**Strengths:**

1. The paper presents ScaleTraversal, a user-friendly and flexible interactive tool for customizing 3D visualizations of multi-scale biomedical data. This tool has the potential to become a powerful aid for domain experts in related fields.
2. The innovative design of the bi-functional user interface strikes a balance between user-friendliness and flexibility. The graphical operations make the interface accessible to non-expert users, while the embedded DSL codes provide fine-grained control.
3. The proposed four memory management strategies for multi-scale data effectively address the challenges of processing large datasets with limited hardware resources. These strategies include streaming data processing, data prefetching, memory utilization optimization, and GPU rendering acceleration.
4. The quantitative experiments and case studies convincingly demonstrate the performance and expressiveness of the tool. The authors have also sought feedback from domain experts, confirming its practicality.

This is an excellent piece of work, and I look forward to seeing your presentation at ACM MM.

**Limitations:**

Although the paper mentions that ScaleTraversal can be extended to other multi-scale scenarios, it does not provide specific methods or examples. This point requires further elaboration and validation.

**Suitability:**

3

---

### Official Review · Reviewer_qTxg · 2024-05-25

**Rating:** 2
**Confidence:** 4

**Summary:**

This paper introduces a  system for interactive 3D demonstration. It also introduces several acceleration strategies, such as  streaming data processing, scale-based data prefetching, dynamic memory deallocation and GPU rendering acceleration. The system runs fluently with limited hardware resources.

**Strengths:**

It seems the system works well and is able to render biomedical data on a machine with limited resources.

**Limitations:**

The paper designs a biomedical data demonstration system. I cannot find any significant novelty in the system. The acceleration strategies are all common practices.

Even though it seems the system works well, this is a system designing work, which is not suitable for the multimedia conference.

**Suitability:**

1

---

### Official Review · Reviewer_MpJE · 2024-05-25

**Rating:** 4
**Confidence:** 2

**Summary:**

The paper presents a visualization tool designed to create animations across data represented at different scale. The tool is nice, and provides a waterfall-like interface to build customized animation representing the different layers

**Strengths:**

The interfaces seems easy to use, some optimization seems to have been implemented to accelerate the rendering. The method seem able to handle a large dataset not visualized by a competitor method. User feedback has been collected from four experts and the feedback is positive.

**Limitations:**

The software is presented as a general tool, but actually it seems designed for a particular kind of dataset and does not seem generalizable. Probably in the description of the design goals also the specificity of the application presented and of the multi scale data used should have been presented.
Are the dataset at the different scales covering the same space? Which are the data included in all the modeled parts? All 3D meshes with attributes?  How is the scale transition practically programmed? As one of the goal of the tool is to enable user to flexibly customize animations, it could be useful to describe in detail the design of the user interface and how the user actually plans the navigation. If they has to write sequences of actions as it seems, it may be not easy to create meaningful animation, so I suppose it would be better to explore and record paths on an interactive navigation tool. But this does not seem available and it is strange. The implementation seem to be able to support this.
As the tool is not interactive, the multi-scale capability seem only related to the fact that when a new animation block at a different scale is added, it is started in the corresponding position, is it right?

**Suitability:**

2

---

### Meta-Review · Senior_Area_Chairs · 2024-06-30

**Recommendation:** Accept (Poster)
**Confidence:** 5

**Metareview:**

3 reviews completed with 1 borderline reject, 1 borderline accept, and 1 accept. Area Chair concurs with majority of reviewers for conditional accept as highest ratest paper in group of papers meta-reviewed by this AC.  Poster Accept remains conditional based on need for 3 revisions: 1) Clarify financial conflict-of-interest if commercial software or alternatively clarify availability of open source software; 2) Clarify and summarize in a table the real-world rendering times to initial display of first appearance of image and animation for the different methods; 3) Clarify history, origin, and/or novelty of each of the design principles especially given the concerns of the reviewers --- please group those design principles that can be found in prior literature and cite previously published literature appropriately, and separately group those design principles that the authors believe to be novel. Summarize both groups of design principles, old and new, in an easily readable text-based table with summary phrases for each design principle.  In order to fit in 8 pages, the authors may have to remove one of the (unnecessary?) pictographic figures that are not really helpful or needed especially if the software is freely available as open source software.